# Novel Universal Recombinant Rotavirus A Vaccine Candidate: Evaluation of Immunological Properties

**DOI:** 10.3390/v16030438

**Published:** 2024-03-12

**Authors:** Dmitriy L. Granovskiy, Nelli S. Khudainazarova, Ekaterina A. Evtushenko, Ekaterina M. Ryabchevskaya, Olga A. Kondakova, Marina V. Arkhipenko, Marina V. Kovrizhko, Elena P. Kolpakova, Tatyana I. Tverdokhlebova, Nikolai A. Nikitin, Olga V. Karpova

**Affiliations:** 1Department of Virology, Faculty of Biology, Lomonosov Moscow State University, 119991 Moscow, Russia; nelly.khudaynazarova@bk.ru (N.S.K.); trifonova.katerina@gmail.com (E.A.E.); eryabchevskaya@gmail.com (E.M.R.); olgakond1@yandex.ru (O.A.K.); armar74@mail.ru (M.V.A.); nikitin@mail.bio.msu.ru (N.A.N.); okar@genebee.msu.ru (O.V.K.); 2Rostov Research Institute of Microbiology and Parasitology, 344010 Rostov-On-Don, Russia; npo-kovrizhko@yandex.ru (M.V.K.); kolpako-va@mail.ru (E.P.K.); rostovniimp@rniimp.ru (T.I.T.)

**Keywords:** rotavirus, rotavirus vaccine, recombinant vaccine, recombinant antigen, structurally modified plant viruses, tobacco mosaic virus, plant virus adjuvants, spherical particles

## Abstract

Rotavirus infection is a leading cause of severe dehydrating gastroenteritis in children under 5 years of age. Although rotavirus-associated mortality has decreased considerably because of the introduction of the worldwide rotavirus vaccination, the global burden of rotavirus-associated gastroenteritis remains high. Current vaccines have a number of disadvantages; therefore, there is a need for innovative approaches in rotavirus vaccine development. In the current study, a universal recombinant rotavirus antigen (URRA) for a novel recombinant vaccine candidate against rotavirus A was obtained and characterised. This antigen included sequences of the VP8* subunit of rotavirus spike protein VP4. For the URRA, for the first time, two approaches were implemented simultaneously—the application of a highly conserved neutralising epitope and the use of the consensus of the extended protein’s fragment. The recognition of URRA by antisera to patient-derived field rotavirus isolates was proven. Plant virus-based spherical particles (SPs), a novel, effective and safe adjuvant, considerably enhanced the immunogenicity of the URRA in a mouse model. Given these facts, a URRA + SPs vaccine candidate is regarded as a prospective basis for a universal vaccine against rotavirus.

## 1. Introduction

Group A rotaviruses (RVA) remain a leading cause of acute gastroenteritis in young children and infants throughout the world. Rotavirus infection is responsible for an estimated 258 million episodes of diarrhoea and 130,000 deaths among children under 5 years of age annually, with a disproportionately high occurrence in low-income countries [1]. The best way to prevent rotavirus infection is vaccination. Since 2006, four attenuated rotavirus vaccines have been licensed in more than 100 countries worldwide. All these vaccines are live-attenuated and require an oral route of administration [2]. Two of these vaccines are used most widely: RotaTeq^®^ (Merck & Co., Rahway, NJ, USA) and Rotarix™ (GlaxoSmithKline, Rixensart, Belgium). RotaTeq^®^ is a live-attenuated pentavalent vaccine based on human–bovine reassortant rotavirus strains (with antigens G1, G2, G3, G4, and P[8]). Rotarix™ is a monovalent live-attenuated vaccine based on the human RVA strain G1P[8]. Both vaccines have been shown to be highly effective in preventing severe rotavirus infection in middle- and high-income countries, but post-licensure studies have demonstrated that existing vaccines have been far less efficacious in low-income countries, where the incidence of rotavirus-associated diarrhoea is already high [3]. Presumably, this is related to the high titres of maternally-derived antibodies, co-infections with other enteropathogens, and the greater diversity of rotavirus circulating in these countries [2]. The disadvantages of existing rotavirus vaccines include some serious side effects, such as intestinal intussusception [4,5,6,7], a wide range of contraindications, the risks of chronic infection [8,9,10], the reversion of the vaccine strain to a virulent phenotype [11], and the recombination of the vaccine strain with wild-type strains [12,13,14]. The latter poses risks of the emergence of new, more pathogenic rotavirus strains [15]. The extent to which existing live rotavirus vaccines provide protection against non-vaccine genotypes is currently a controversial issue. Several studies reported substantial changes in the composition of circulating RVA strains and an increase in the proportion of heterotypic genotypes in the post-vaccination era [16,17]. However, it is not yet known whether these changes are due to the selective pressure of vaccines or natural evolutionary processes [18,19].

The shortcomings of the existing vaccines highlight the need for newer approaches to rotavirus vaccine development. One of the most promising directions of research in this field is the creation of non-replicating recombinant vaccines with a parenteral route of administration. The use of such vaccines avoids the multiplication of the vaccine strain in the intestine, which minimises the risk of side effects, chronic infection, and reassortment with wild-type RVA. The advantages of these vaccines also include greater safety and purity of the preparation [20,21]. For the development of a recombinant RVA vaccine, it is advisable to use rotavirus structural proteins, which are essential for an effective immune response. The rotavirus virion is composed of three protein shells—an outer capsid, an inner capsid, and an internal core—that enclose 11 segments of double-stranded RNA. Two rotaviral structural proteins of the outer capsid, the spike protein VP4 (protease-cleaved protein, P), and VP7 (glycosylated protein, G), define both the serotype and the genotype of rotavirus strains [22] and are considered to be crucial for vaccine development [23,24]. During infection, the VP4 spike protein is cleaved by intestinal trypsin into two subunits: VP8* and VP5*. Both of them provide a good basis for the development of a recombinant rotavirus vaccine [25,26,27,28,29]. In particular, it has been shown that a VP8*-induced immune response is sufficient for disease prevention [30]. However, studies have revealed that such subunit vaccine candidates generate low cross-reactive immune responses to heterologous strains of RVA [27]. In an attempt to provide broader protection, multivalent vaccines based on antigens from several RVA genotypes have been formulated [31], but further evaluations of efficiency are required to better understand the ability of such vaccines to provide cross-serotype protection.

Peptides corresponding to the neutralising epitopes of rotavirus antigens are also being considered as a potential basis for a recombinant vaccine against rotavirus. It is assumed that the use of peptides that mimic pathogen’s epitopes allows the production of antibodies against specific regions of antigens, including sites that may be otherwise inaccessible to the immune system [32,33]. In addition, it is assumed that immunisation with constructs containing highly conserved epitopes induces the production of cross-reactive antibodies, which is essential since strain diversity is one of the most fundamental problems in the development of a vaccine against RVA [34]. Despite such possibilities, the development of peptide-based vaccines remains limited. This is most likely related to the poor immunogenic activity of peptides themselves and the lack of effective and safe adjuvants available for use in humans [32].

The present work is devoted to the development of a novel, broad-spectrum, highly immunogenic RVA vaccine candidate. A new recombinant rotavirus antigen was designed as the basis for the vaccine candidate. For this, two methods were combined: the application of a conserved neutralising epitope and the use of the consensus of the extended protein’s fragment. The resulting antigen, named URRA (Universal Recombinant Rotavirus Antigen), consists of a short peptide ep8 corresponding to a neutralising epitope (from 1 to 10 aa VP8*) highly conserved among RVA strains [34,35] and ΔVP8*, the truncated VP8* subunit (from 65 to 223 aa of VP8*), obtained using the consensus approach on the base of a wide range of RVA isolates of genotype P[8] [28]. The coding sequences of ep8 and ΔVP8* were designed previously [28]. The potential impact of vaccines’ selective pressure on RVA strains’ genetic diversity and distribution, the emergence of previously uncommon RVA strains, and rapid evolutionary changes in the RVA population are all concerns that are widely discussed [14,16,17,18,19,36,37]. Because of this, the current research focused on the antigenic properties of the URRA protein. The correspondence of URRA to currently circulating RVA variants was examined using antisera to patient-derived field RVA isolates.

The proposed vaccine candidate contains a special adjuvant, spherical particles (SPs) obtained from the tobacco mosaic virus (TMV). Previous studies have demonstrated the properties of TMV SPs as an adjuvant and a platform for the adsorption and stabilisation of various antigens [38,39,40]. SPs are also known to be safe in a wide variety of animal models and biodegradable [41,42,43]. Here, the immunogenicity of the vaccine candidate was evaluated in a murine model and compared with the immunogenicity of the individually formulated URRA. The immunogenicity of SPs was measured separately to estimate the immune response to the adjuvant. The data obtained show that a vaccine candidate based on a URRA + SPs composition provides a possible solution to the fundamental challenges of recombinant RVA vaccine development. Such a vaccine could be a prospective subject for further research.

## 2. Materials and Methods

### 2.1. Expression and Purification of Rotavirus Recombinant Antigen

*Escherichia coli* strain XL1-Blue was applied for rotavirus recombinant antigen expression. The cultures were grown in 3 mL of a 2YT medium containing 1.6% (*w*/*v*) tryptone, 1% (*w*/*v*) yeast extract, 0.5% (*w*/*v*) NaCl, and 100 µg/mL of ampicillin at a temperature of 37 °C with shaking at 180 rpm overnight. The cultures were added to 200 mL of 2YT with the same composition and were grown at 37 °C with shaking at 180 rpm for 3 h. After that, cultures were induced with IPTG to a final concentration of 2 mM and were cultured for 5 h at 37 °C and shaking at 180 rpm. Cell pellets were centrifugated for 10 min at 5000× *g* (JA-14 rotor, Avanti JXN-30 centrifuge, Beckman Coulter Inc., Brea, CA, USA) at 4 °C, then stored at −20 °C and subsequently used for chromatographic isolation and the purification of recombinant protein. Metal affinity chromatography with Ni^2+^-NTA resin (Qiagen, Hilden, Germany) under denaturing conditions was applied. For this, sediment cell pellets were resuspended and lysed in 5 mL of a solution containing 6 M GuHCl and 0.2% (*w*/*v*) natrium deoxycholate at 25 °C with shaking at 120 rpm for 1 h. The recombinant protein was eluted from the column according to the manufacturer’s protocol (Qiagen) and then dialysed against deionised water (for 2 h) and Milli-Q (for 2 h) (Simplicity UV, Merck Millipore, Darmstadt, Germany) in the ratio 1:250, with hourly water replacement, and then stored at −20 °C.

### 2.2. Obtaining Sera to Untyped Field Rotavirus Isolate

Serum №1 and Serum №2 were obtained from two corresponding groups of outbreed CD-1 mice. Each group consisted of 10 individual males aged 6–8 weeks old. Mice were immunised with RV strains RVV-5 (Serum №1) and RRV-6 (Serum №2). These strains are stored in the collection of the Rostov Research Institute of Microbiology and Parasitology (Rostov-On-Don, Russian Federation). The strains were isolated from the material obtained from children with laboratory-confirmed rotavirus gastroenteritis who were undergoing hospital treatment in the infectious diseases department of the City Hospital of Rostov-on-Don. The bacterial and fungal flora-free material was adapted to growth on continuous mammalian cell cultures VERO and SPEV and purified by high-speed centrifugation. Both strains were assigned to RVA by PCR and ELISA using appropriate test systems (manufactured by Vector-Best, Russian Federation; AmpliSens, Russian Federation). Mice were immunised twice intramuscularly in two pelvic limbs in equal amounts (100 μL in each limb). No adjuvants were used for the immunisation.

### 2.3. Western Blot Analysis

First, an SDS-PAGE with an 8–20% acrylamide linear gradient was performed. Proteins separated by electrophoresis were then transferred to a PVDF membrane (Invitrogen, TM, ThermoFisher Scientific, Waltham, MA, USA) using a MINI PROTEAN II (Bio-Rad Laboratories Inc., USA) transfer system. The membrane was blocked with 5% (*w*/*v*) non-fat dry milk in TTBS (0.01 M Tris-HCl (pH 7.4), 0.15 M NaCl, and 0.05% (*v*/*v*) Tween-20). Then, the membrane was treated with primary mice polyclonal Abs (Serum №1 and Serum №2; a description is given in Section 2.2 of the “Materials and Methods”) to untyped patient-derived rotavirus isolates in a 1:500 dilution, and then with secondary Abs to mouse IgG conjugated with horseradish peroxidase (Jackson Immunoresearch Inc., West Grove, PA, USA) in a 1:20,000 dilution. WesternBright ECL substrate (Advansta Inc., San Jose, CA, USA) was applied, and the signal was detected using the ChemiDoc™ XRS documentation system with Image Lab™ Software Version 6.1 (Bio-Rad Laboratories, Hercules, CA, USA).

### 2.4. Enzyme-Linked Immunosorbent Assay (ELISA) for Qualitative Assessment of Protein-Serum Interaction

For the qualitative assessment of the interaction of URRA with Serum №1 and Serum №2, incubation and washing schedules were consistent with the protocol described by Kovalenko et al. (2022) [39]. URRA or SPs were used as an antigen for coating a 96-well plate in concentrations of 10 μg/mL, 50 μg/mL, 100 μg/mL, or 200 μg/mL. Analyses were conducted in two replicates for each antigen concentration for each serum analysed. Serum №1 or Serum №2 were used in a dilution of 1:100. Anti-mouse total IgG HRP conjugate (#ab6728, Abcam, Cambridge, UK) was used in a dilution of 1:10,000.

### 2.5. TMV Isolation and Spherical Particles Generation

The TMV and the SPs were obtained according to the protocol described by Trifonova et al. (2015) [44], with some modifications. A TMV solution with a concentration of 2 mg/mL was used for SPs formation. The TMV solution was aliquoted into 1.5 mL polypropylene tubes (Greiner Bio-One GmbH, Frickenhausen, Germany). Aliquots of 500 μL each were heated to 98 °C in a “Termite” thermostat (DNA technology, Moscow, Russia) and incubated at 98 °C for 10 min. After cooling the aliquots at 4 °C for 5 min, the preparations were vortexed and re-incubated at 98 °C for 10 min.

### 2.6. Immunofluorescence Analysis

URRA + SPs or SPs samples formulated in PBS were loaded onto the coverslips coated with formvar. The loaded samples were incubated for 10 min. Then, the excess of the samples was removed. Then, the coverslips were dried in the air for 10 min. The resulting coverslips with the loaded samples were incubated for 1 h with a blocking solution (1% bovine serum albumin (BSA) and 0.05% Tween-20 in PBS); then, for 1 h with 1:50 dilution of polyclonal anti-URRA serum obtained from mice immunised with URRA twice with a two-week interval between immunisations. During the first immunisation, complete Freund’s adjuvant was applied, while during the second, the incomplete version was used. For controls without primary antibodies, the coverslips were incubated with a blocking solution for an additional hour instead. The coverslips were washed three times with a washing solution (0.25% BSA and 0.05% Tween-20 in PBS) and subsequently incubated for 1 h with Alexa 546 fluorophore-conjugated secondary antibodies to mouse IgG (Invitrogen, USA; 1:100 dilution in the blocking solution). After that, the coverslips were washed three times with the washing solution, once with PBS, and finally rinsed with pure water and dried in air. Immediately prior to the examination of the samples, the preparations were treated with a photo-protector 1,4-diazabicyclo[2.2.2]octane and studied under an Axiovert 200 M fluorescence microscope (Carl Zeiss, Oberkochen, Germany) equipped with an ORCAII-ERG2 integrated camera (Hamamatsu Photonics, Shizuoka, Japan).

### 2.7. Immunisation of Mice for the Immunogenicity Studies

To study the immunogenicity of the vaccine candidate and the individually formulated URRA, four groups of BALB/c mice were used. Each group consisted of 25 individual females. The non-immunised control was represented by group 1. Group 2 served as an adjuvant control, and the mice were immunised with SPs and TMV in an amount of 250 μg per dose. Group 3 mice were immunised with individual rotavirus antigen URRA in an amount of 15 μg per dose. Those in Group 4 were immunised with a URRA + SPs composition. For this group, one dose contained 15 μg of URRA protein and 250 μg of SPs; therefore, the ratio of antigen to SPs, by mass, was 15:250. A description of the immunisation groups and the scheme of the experiment are provided in Section 3.4 of the “Results”. All samples administered were prepared in PBS. The final volume of one dose was 260 μL/animal. The mice were immunised intramuscularly in a pelvic limb with 260 μL of the solution. The pelvic limb selected for the immunisation was changed between the immunisations. Blood collection and euthanasia were carried out by decapitation [45].

### 2.8. Ethical Statement

The immunogenicity evaluation experiments on mice were approved by the Ethics Committee of the Sechenov First Moscow State Medical University (Protocol №111 dated 21 October 2022). The experiments for obtaining Serum №1 and Serum №2 were approved by the Ethics Committee of the Rostov Scientific Research Institute of Microbiology and Parasitology (Protocol №05/17 dated 23 May 2023).

### 2.9. Statistical Analysis

The Mann–Whitney Test with the Holm–Bonferroni correction was used for multiple comparisons. The Mann–Whitney Test was used for a single pairwise comparison. Comparison results were considered to be significant with a probability value (*p*-value) of less than 0.05. Statistical processing of the results and the plotting of graphs were carried out using the GraphPadPrism 9.1.0 program (GraphPad Software, La Jolla, San Diego, CA, USA).

### 2.10. Enzyme-Linked Immunosorbent Assay (ELISA) for Titre Measurement

For the measurement of anti-URRA and anti-SPs antibody titres, an ELISA was performed according to the protocol described by Kovalenko et al. (2022) [39]. URRA or SPs were used as an antigen for coating the 96-well plates with a concentration of 10 μg/mL. All sera samples collected were titrated in three-fold serial dilutions, starting from 1:30. Anti-mouse total IgG HRP conjugate (#ab6728, Abcam, Cambridge, UK), anti-mouse IgG1 HRP conjugate (#ab97240, Abcam, Cambridge, UK), anti-mouse IgG2a HRP conjugate (#ab97245, Abcam, Cambridge, UK), anti-mouse IgG2b HRP conjugate (#ab97250, Abcam, Cambridge, UK), or anti-mouse total IgG3 HRP conjugate (#ab97260, Abcam, Cambridge, UK) was used in a dilution of 1:10,000. The serum titre was defined as the reciprocal of the serum dilution at which A_450_ was equal to the mean of the background signal + 3 SD. The background signal was taken as the mean value obtained from 24 wells for each plate separately, into which no test serum was added (neither experimental nor non-immune). If the A_450_ in a 1:30 dilution was below the mean value of the background signal + 3 SD, the serum titre was considered to be 30. If the sera titre was measured in more than one replicate, the geometric mean of these values was used for further calculations and presentation.

## 3. Results

### 3.1. Designing Universal Recombinant Rotavirus Antigen URRA

A Universal recombinant rotavirus antigen (URRA) was designed for the purpose of the current research (Figure 1). The amino acid sequence of the URRA is based on the sequence of the rotavirus VP8* protein (one of two subunits formed following rotavirus spike protein VP4 cleavage by trypsin). It consists of two parts. The N-terminus of the protein is represented by a short peptide ep8 corresponding to a linear neutralising B cell epitope with the sequence ^1^MASLIYRQLL^10^ (1–10 aa of VP8*), which is highly conserved among the vast majority of RVA strains of all genotypes. The ep8 peptide sequence is followed by the sequence of the ΔVP8*P[8] towards the C-terminus of the protein. The ΔVP8*P[8] is the truncated VP8* subunit (65–223 aa of VP8*) obtained using a consensus approach based on a wide range of RVA isolates of genotype P[8], as described previously [28]. The C-terminus of the URRA contains a hexahistidine tag-coding sequence. The estimated molecular weight of the URRA, calculated by amino acid sequence using the ProtParam EXPaSy proteomics server, Swiss Institute of Bioinformatics, http://expasy.org/ (accessed on 6 February 2024), was 20.397 kDa.

### 3.2. Interaction of URRA with Antisera to Patient-Derived Field Rotavirus Isolates

The ability of the recombinant rotavirus antigen, URRA, to interact with polyclonal antisera to untyped patient-derived field rotavirus isolates circulating in the Russian Federation was evaluated. Serum №1 and Serum №2, described in Section 2.2 of the “Materials and Methods”, were used for the analyses. The interaction of the URRA with both antisera was qualitatively assessed by means of an ELISA, as described in Section 2.4 of the “Materials and Methods”, and a Western blot analysis. The results of the ELISA and Western blot analysis performed using Serum №1 and the ELISA performed using Serum №2 are presented in Figure 2, Figure 3 and Figure 4, respectively. For Serum №1, the recognition of the URRA by the serum was demonstrated by both the ELISA (Figure 2, Appendix A) and the Western blot analysis (Figure 3a, lane 3). For Serum №2, the recognition of the URRA by the serum was demonstrated by the ELISA (Figure 4, Appendix A), but the Western blot analysis did not reveal any interaction between the antigen and the serum.

### 3.3. The Adsorption of URRA to SPs

In this study, the authors propose the use of spherical particles (SPs), obtained from the tobacco mosaic virus through heat treatment, as an adjuvant for the URRA in a vaccine candidate formulation. The ability of the URRA to form composition with SPs by adsorbing to their surface was examined by indirect immunofluorescence analysis (Figure 5. Negative controls are presented in Appendix A). The URRA:SPs mass ratio within the composition was 15:250. The results obtained demonstrated that the URRA was able to effectively adsorb to SPs. The presence of the fluorescent signal indicates that the rotavirus antigen URRA maintains its antigenic properties while being adsorbed to SPs.

### 3.4. The Immunogenicity of Individual URRA and of a Vaccine Candidate (URRA + SPs)

The immunisation of the mice was carried out to evaluate the immunogenicity of the vaccine candidate and the impact of SPs on the immunogenicity of the URRA. The immunisation schedule and a brief description of the groups of mice are presented in Figure 6. Four groups of mice, each consisting of 25 animals, were used in the experiment. Group 1 was not immunised and served as a control group. Groups 2, 3, and 4 were immunised with SPs (which served as an adjuvant control), URRA, or the URRA + SPs composition, respectively. For all groups, ten mice were immunised once, and blood was collected on the 21^st^ day after immunisation; 15 mice were immunised twice, with a 21 day interval between immunisations, and the blood was collected on the 42^nd^ day after the second immunisation.

The sera were obtained from all blood samples collected. Total anti-URRA IgG titres were measured for all sera samples obtained using an ELISA, as described in Section 2.10 of the “Materials and Methods”. The results of the ELISA and statistical analyses carried out for sera from mice after the first immunisation are presented in Figure 7 (complete data on titres are presented in Appendix A). Those for sera obtained after the second immunisation are presented in Figure 8 (complete data on titres are presented in Appendix A). The Wilcoxon–Mann–Whitney Test with the Holm–Bonferroni correction was used for subsequent comparisons for sera obtained from both once- and twice-immunised mice. Anti-URRA total IgG titres elicited by the individually formulated URRA (group 3) or the URRA + SPs composition (group 4, vaccine candidate) were subjected to pairwise comparison with those in the non-immunised group (group 1). Anti-URRA IgG titres elicited by the individual URRA and the URRA + SPs composition were compared with each other to evaluate the impact of SPs on the immunogenicity of the URRA. Finally, titres from groups immunised with SPs-containing formulations (groups 2 and 4) were also compared with each other.

After the first immunisation (Figure 7, Appendix A), of all the pairwise comparisons conducted, significant differences in anti-URRA IgG titres were revealed between sera from the group immunised with the URRA + SPs composition (group 4, median titre 4.5 × 10^2^) and two control groups: the non-immunised group (group 1, median titre 6.94 × 10^1^) and the group immunised with SPs (group 2, adjuvant control, median titre 6.73 × 10^1^). There was no significant difference between anti-URRA IgG titres of mice immunised with the individual antigen URRA (group 3, median titre 1.59 × 10^2^) and either the sera titres of mice from the non-immunised group (group 1) or those from the group immunised with the vaccine candidate (group 4).

After the second immunisation (Figure 8, Appendix A), the significant difference found in anti-URRA IgG titres between the non-immunised control group (group 1, median titre 2.05 × 10^2^) and the group immunised with the vaccine candidate (group 4, median titre 2.18 × 10^4^) was again demonstrated. In contrast to sera obtained after the first immunisation, a significant difference was demonstrated between anti-URRA IgG titres induced by the URRA (group 3, median titre 1.99 × 10^3^) and those induced by the vaccine candidate, which were 11 times higher. The significant difference between the groups immunised with SPs only (group 2, median titre 2.03 × 10^2^) and those immunised with the URRA + SPs composition was also repeated.

In the present study, the immunogenicity of the vaccine candidate and the individual URRA after the second immunisation was assessed not only by total IgG titres but also separately by IgG1, IgG2a, IgG2b, and IgG3 isotype titres. The ELISA and statistical analyses were carried out in the same manner as they were for the assessment of total IgG titres. The results of the ELISA and statistical analyses are presented in Figure 9 (the complete data on anti-URRA IgG1, IgG2a, IgG2b, and IgG3 sera titres are presented in Appendix A, respectively). For IgG2a, IgG2b, and IgG3, no significant differences were revealed between any of the groups compared. Both the individual URRA (group 3, median titre 1.5 × 10^3^) and the vaccine candidate (group 4, median titre 4.51 × 10^4^) elicited a significant number of anti-URRA IgG1 antibodies, compared with the non-immunised control group (group 1, median titre 8.65 × 10^1^). The sera IgG1 titres induced by the vaccine candidate were 30 times higher than those induced by the individual URRA. The significant difference in anti-URRA IgG1 titres was also revealed between the group immunised with the vaccine candidate and the group immunised with SPs only (group 2, median titre 1.46 × 10^2^).

The ratio of the immune response to an antigen and an adjuvant is an important characteristic of a vaccine. Thus, the anti-URRA and anti-SPs total IgG titres elicited after two immunisations with the vaccine candidate were measured with an ELISA and compared using the Wilcoxon–Mann–Whitney Test. The results are presented in Figure 10 (the complete data on titres are presented in Appendix A). It was revealed that the titres of anti-URRA IgG (median titre 2.18 × 10^4^) were 14 times higher than those of anti-SPs IgG (median titre 1.52 × 10^3^).

## 4. Discussion

In the present work, a recombinant rotavirus A (RVA) antigen was developed based on the VP8* subunit of the VP4 spike protein. VP4 is one of the components of the rotavirus capsid’s outer layer. During infection, VP4 is cleaved by intestinal trypsin into two subunits, VP8* and VP5*, which are rotavirus antigens containing various neutralising epitopes. Here, to obtain a universal RVA antigen URRA, two approaches were combined—the application of the extended protein’s fragment and the use of a conserved neutralising epitope. Thus, the antigen URRA is composed of two parts. The larger part is ΔVP8*P[8], the truncated VP8* subunit (65–223 aa of VP8*) obtained using a consensus approach on the base of a wide range of RVA isolates of genotype P[8]. The coding sequence of ΔVP8*P[8] was designed as described previously [28]. The consensus sequence of ΔVP8*P[8] represents the “spike head”, the concanavalin-like domain of the VP8* subunit, and contains 159 amino acid residues. Both VP8* and its truncated forms are known to be highly immunogenic and stimulate the production of virus-neutralising antibodies, which makes VP8* a promising basis for a vaccine candidate against RVA [27,31]. One of the most advanced developments in this area is the trivalent subunit vaccine P2-VP8* [31]. It consists of a truncated VP8* segment (65–223 aa) of the three most common RVA genotypes—P[8], P[4], and P[6]—fused to the Th2 epitope of the tetanus toxoid to enhance immunogenicity. This vaccine candidate has successfully passed phase 1 and 2 clinical trials and is currently in phase 3 (ClinicalTrials.gov NCT04010488). The vaccine candidate P2-VP8* was shown to induce a high immune response and elicit the production of neutralising antibodies to the P[4], P[6], and P[8] genotypes of RVA strains [31]. At the same time, 58 P-genotypes of rotavirus have already been described in humans and animals worldwide [46], and more research is required to evaluate the ability of subunit vaccines to provide sufficiently broad cross-serotype protection. A promising approach to extending the vaccine’s protection is the application of short peptides that mimic neutralising epitopes, highly conserved among a wide range of RVA strains. For this reason, the lesser, N-terminal part of the URRA is represented by the short peptide ep8 corresponding to a linear neutralising B-cell epitope with the sequence ^1^MASLIYRQLL^10^ (1–10 aa of VP8*) [34,35], which is highly conserved among the vast majority of RV strains of all genotypes. The sequence of ep8 was obtained as described previously [28]. Peptide vaccines are supposed to induce specific immune responses to pathogens’ neutralising epitopes, including those epitopes that are otherwise inaccessible to the immune system. Conversely, owing to the relatively small size of peptides, they are often weakly immunogenic and, therefore, require carrier platforms for delivery and adjuvating [32,33]. In the study by Kovacs-Nolan et al. (2006), the immunogenicity of a peptide with the sequence ^1^MASLIYRQLL^10^ (1–10 aa VP8*) was evaluated in an animal model [34]. The peptide was covalently linked to a thioredoxin carrier protein fused with the P2 epitope of the tetanus toxin in order to boost immunogenicity. The authors of this paper believe that immunisation with such constructs has the potential to induce broad-spectrum immunity against multiple serological variants of RVA.

The global introduction of rotavirus vaccines has resulted in a considerable reduction in rotavirus-related deaths and hospitalisations. However, recent studies revealed changes in the composition of circulating RVA strains and the emergence of previously uncommon rotavirus strains after the introduction of the vaccines [14,16,17,47]. This may be due to either vaccine selective pressure or natural evolutionary processes [18,19]. One way or another, this situation raises concerns and highlights the necessity to focus on currently relevant RVA strains when developing a new vaccine. For this reason, the authors decided to examine the antigenic specificity of the URRA protein using polyclonal antisera (Serum №1 and Serum №2) to patient-derived field RVA isolates currently circulating in the Russian Federation. The possibility of the URRA interacting with these sera was proven using the Western blot and indirect ELISA analyses. In the ELISA assay, Serum №1 interacted effectively with the URRA (Figure 2). The Western blot analysis performed with Serum №1 also revealed the recognition of the URRA (Figure 3a, lane 3). Thus, the results of the Western blot and ELISA analyses with Serum №1 coincided, revealing that the URRA corresponds to a relevant RVA field isolate. Serum №2 interacted effectively with the URRA in the ELISA assay (Figure 4). However, no recognition of the antigen in the Western blot analysis was observed. In this case, the results of the Western blot analysis and ELISA are contradictory. This may be explained by the fact that Serum №2 recognised some conformational epitopes in the URRA, which are well-preserved under the native conditions of an ELISA but are not preserved under the denaturing conditions of the Western blot analysis. Certainly, the authors’ assumption requires further research. Nevertheless, it is concluded that the results of the experiments indicate the ability of the URRA to interact with both antisera and currently circulating field RVA isolates.

The authors proposed a vaccine candidate, URRA + SPs, that represents the composition of URRA and spherical particles (SPs) generated by the thermally-induced rearrangement of the tobacco mosaic virus (TMV). Previous studies have demonstrated the properties of TMV SPs as a prospective safe and biodegradable adjuvant and a platform for the adsorption and stabilisation of various antigens [38,39,40,41,42,43,44]. To examine the antigenic properties of the URRA in the URRA + SPs composition, immunofluorescence analysis with primary polyclonal sera to the URRA was carried out, revealing that the URRA adsorbs effectively to SPs while maintaining antigenic specificity. This suggests that SPs could be used as a platform for the URRA-based vaccine candidate. The immunogenicity of the vaccine candidate was evaluated and compared with the immunogenicity of the individual URRA. The experiments were carried out in mice. After the first immunisation, the anti-URRA total IgG titres induced by the URRA + SPs composition were significantly higher than the titres elicited in control groups. In contrast, the differences between anti-URRA IgG titres elicited by individual URRA and the titres elicited in control groups were not significant. This indicates the low immunogenicity of the URRA by itself and points to the necessity of using adjuvants in general and SPs as an adjuvant in particular. After the second immunisation, it was revealed that anti-URRA IgG titres elicited by the URRA + SPs were significantly higher than those elicited by the URRA individually. This indicates that SPs increase the immunogenicity of the URRA considerably when co-administered and that SPs could serve as an appropriate adjuvant for the URRA in the vaccine candidate. Anti-URRA IgG titres induced by vaccine candidate, URRA + SPs, after the second immunisation were 48 times higher than anti-URRA titres in the corresponding group after the first immunisation. The results have revealed that at least double immunisation with the vaccine candidate URRA + SPs is required to provide high immunogenicity. Experiments studying the titres of IgG isotypes separately demonstrated that immune responses to both the vaccine candidate and individual antigens are mostly represented by the IgG1 isotype. This indicates that SPs do not alter the polarisation of the immune response to rotavirus antigen URRA. According to various estimates, a natural RVA infection in mice (EDIM, epizootic diarrhoea of infant mice) induces an IgG1 predominant [48], or IgG1/IgG2 balanced [49], immune response.

Protein adjuvants and carriers are known to be able to activate a self-immune response. In the current study, the immunogenicity of SPs was evaluated after double immunisation with the URRA + SPs vaccine candidate. Total IgG titres to SPs were shown to be 14 times less than to the URRA after the second immunisation with the URRA + SPs composition. The prevalence of anti-URRA antibodies confirms the possibility of applying SPs as an adjuvant for a rotavirus vaccine candidate. These results are consistent with previous studies on SPs-based vaccines against rubella, COVID-19, and anthrax [38,39,40]. In all these cases, IgG titres to SPs were significantly lower than to the antigen of interest. A possible limitation of using URRA + SPs as a vaccine composition is the immunity to SPs adjuvant, which is induced during the first immunisation and can potentially reduce the effectiveness of each subsequent one. Pre-existing immunity to a platform or an adjuvant is a serious issue in the vaccine research field. However, it is known that the problem with the effectiveness of booster immunisations does not always arise. Some studies on protein immunopreparations based on plant viruses demonstrated that pre-existing immunity did not reduce the effectiveness of such drugs [50,51], or even increased it [52]. In the current research, a considerable increase in anti-URRA antibody titres was detected after the second immunisation with the URRA + SPs composition compared to the first immunisation (Appendix A, respectively). Presumably, these data indicate that in the case of SPs, the antibodies to a platform might not affect the effectiveness of further immunisations.

## 5. Conclusions

In this research, a sequence of the universal recombinant rotavirus antigen, URRA, was designed based on a wide variety of rotavirus strains, combining two approaches to achieve the goal of creating an antigen able to provide an effective immune response.

In serological studies, URRA demonstrated consistency with rotavirus strains circulating in the Russian Federation. This makes this protein a promising basis for a recombinant rotavirus vaccine. The immunogenicity of the URRA was assessed in individual form and when combined with spherical particles (SPs) obtained from the tobacco mosaic virus. Individual URRA was only able to elicit anti-URRA titres after two immunisations. At the same time, when paired with SPs, the URRA induced a significant immune response, even after a single immunisation. Moreover, SPs were able to enhance the immunogenicity of the URRA after two immunisations. Combined with the fact that the immune response to spherical particles themselves was shown to be significantly lower than that to the rotavirus antigen, the results obtained indicate that they may be considered an appropriate adjuvant for the URRA. Therefore, the recombinant rotavirus antigen, URRA, paired with spherical particles obtained from the tobacco mosaic virus in a 15:250 mass ratio might provide an elegant solution to the challenge of recombinant rotavirus vaccine development.

## 6. Limitations of the Current Study

The present study mainly focuses on the assessment of the immunological properties of the vaccine candidate, including the ability to interact with the antisera to existing rotavirus strains and immunogenicity. In this regard, this research has several strengths and limitations. The main strengths of the study are the demonstrated ability of the URRA to interact with two patient-derived strains of rotavirus circulating in the Russian Federation and the detailed analyses of the total IgG and all IgG subclasses titres elicited after the two-step immunisation of mice with the individual URRA and with a vaccine candidate. In terms of limitations, first and foremost, the immunogenicity data has to be additionally supported by the protectiveness assessment, which is the subject of further investigation. Secondly, comparing the adjuvant effect of SPs on the immunogenicity of the URRA to the effect of other adjuvants might provide an overall picture for the further rotavirus vaccine design. Finally, the assessment of IgA titres in the vaccinated mice may enable us to draw deeper conclusions about the effectiveness of the vaccine candidate.

## Figures and Tables

**Figure 1 viruses-16-00438-f001:**
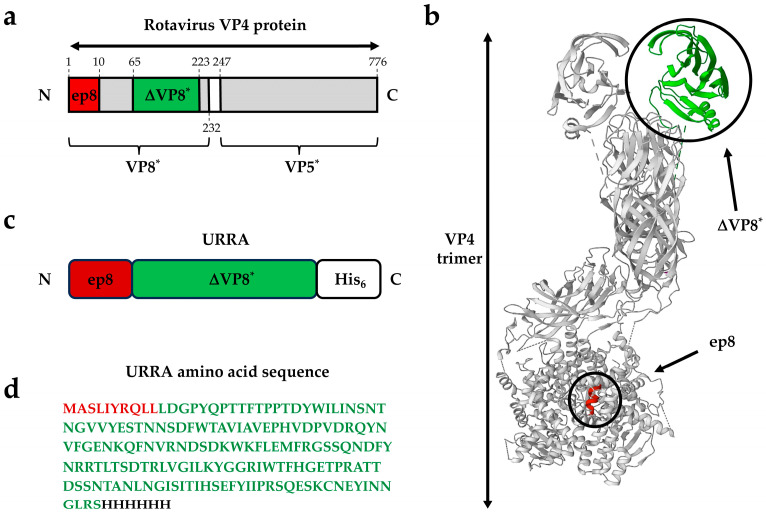
Rotavirus spike protein structure and graphical overview of a URRA. (**a**) Linear diagram of rotavirus VP4 spike protein (1–776 aa) showing the location of fragments VP8* (1–232 aa), VP5* (247–776 aa), ΔVP8* (65–223 aa), epitope ep8 (1–10 aa) (not to scale). VP4 fragment 232–247 aa is being removed during VP4 in vivo proteolysis. (**b**) Structural model of the rotavirus VP4 protein trimer showing fragment ΔVP8* and epitope ep8. The protein VP4 structure was visualised using Mol*, https://molstar.org/ (accessed on 6 February 2024). based on cryo-electron microscopy data (Protein Data Bank [PDB]: 6WXE). (**c**) Schematic representation of the URRA’s structure. (**d**) The amino acid sequence of a URRA. All images (**a**–**d**) use the following colour code: ΔVP8*, green; epitope ep8, red.

**Figure 2 viruses-16-00438-f002:**
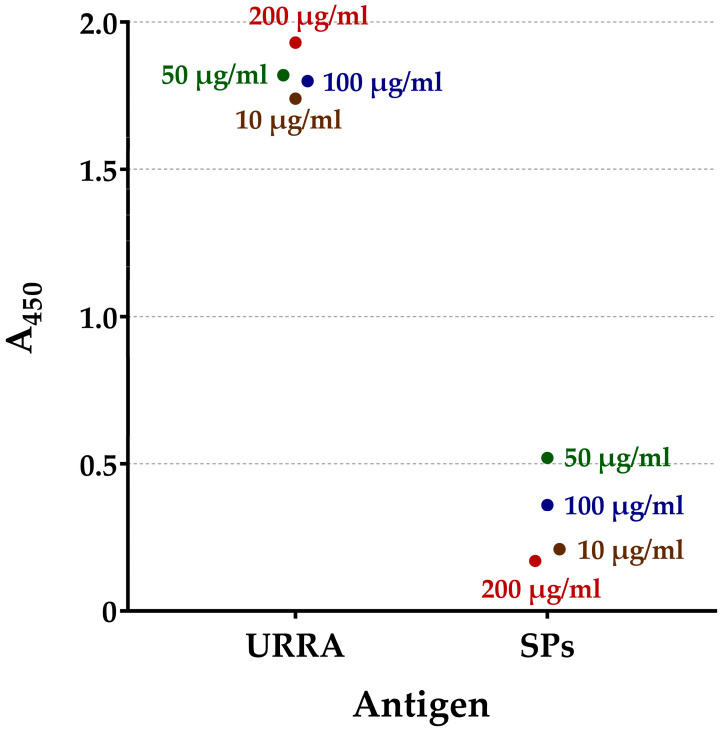
Interaction of URRA with mice antiserum to field rotavirus isolate (Serum №1). The efficiency of interaction was estimated based on absorbance at wavelength 450 nm (A_450_), as evaluated by indirect ELISA. A parallel experiment with SPs as an antigen was performed as a negative control. Secondary HRP-conjugated antibodies to mouse IgG were used. The analyses were carried out using four different concentrations of antigen for adsorption on a microplate. Antigen concentrations are marked on the figure near the corresponding point (10 μg/mL, 50 μg/mL, 100 μg/mL, and 200 μg/mL, respectively). ●, geometric means of A_450_ values for certain antigen concentrations. The analyses were conducted in two replicates for each antigen concentration. The complete data on A_450_, for each replicate and for all analyses, are presented in Appendix A.

**Figure 3 viruses-16-00438-f003:**
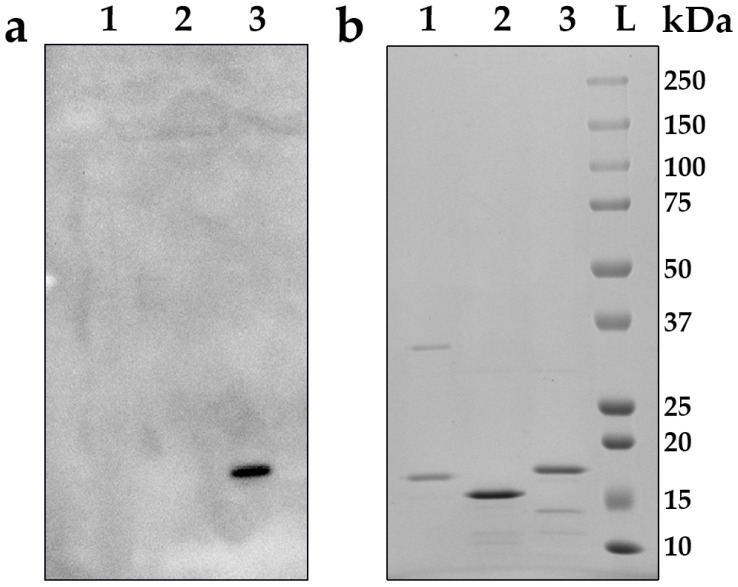
Interaction of URRA with antiserum to field rotavirus isolate (Serum №1). (**a**) Western blot analysis with primary antiserum №1 to field rotavirus isolate (1:500) and secondary HRP-conjugated antibodies (1:10,000). 1—SPs (negative control), 2—heterologous hexahistidine tag-containing recombinant protein (negative control), 3—URRA. (**b**) Electrophoresis analysis in 8–20% SDS-PAGE, staining by Coomassie G-250. 1—SPs (negative control), 2—heterologous hexahistidine tag-containing recombinant protein (negative control), 3—URRA, L—protein molecular weight markers ladder (molecular weights, in kDa, are indicated on the right).

**Figure 4 viruses-16-00438-f004:**
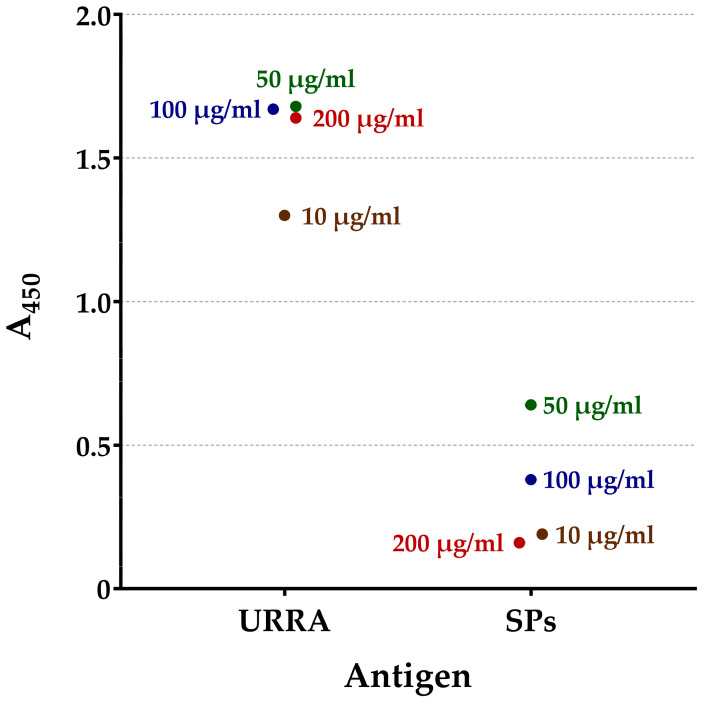
Interaction of URRA with mice antiserum to field rotavirus isolate (Serum №2). The efficiency of interaction was estimated based on absorbance at wavelength 450 nm (A_450_), as evaluated by indirect ELISA. A parallel experiment with SPs as an antigen was performed as a negative control. Secondary HRP-conjugated antibodies to mouse IgG were used. The analyses were carried out using four different concentrations of antigen for adsorption on a microplate. Antigen concentrations are marked on the figure near the corresponding point (10 μg/mL, 50 μg/mL, 100 μg/mL, and 200 μg/mL, respectively). ●, geometric means of A_450_ values for certain antigen concentrations. The analyses were conducted in two replicates for each antigen concentration. The complete data on A_450_, for each replicate and for all analyses, are presented in Appendix A.

**Figure 5 viruses-16-00438-f005:**
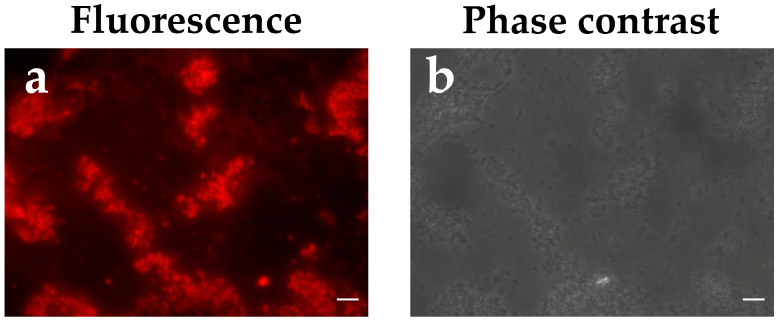
Immunofluorescence analysis of the URRA + SPs composition. (**a**,**b**) are the same image, presented in fluorescence and phase contrast modes, respectively. The URRA + SPs composition was obtained in PBS. The URRA:SPs mass ratio within the composition was 15:250. The URRA + SPs composition was treated with polyclonal mouse anti-URRA serum, obtained using Freund’s adjuvants and secondary antibodies conjugated to Alexa Fluor^®^ 546. Scale bars, 5 μm. Negative controls are presented in Appendix A.

**Figure 6 viruses-16-00438-f006:**
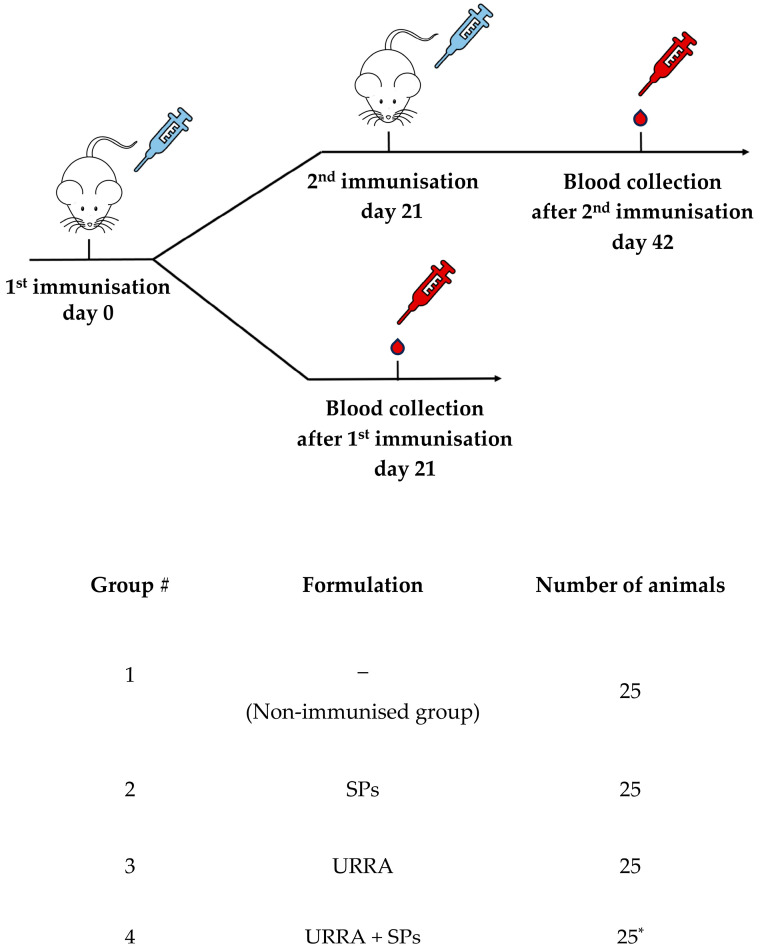
Immunisation schedule and description of mice groups involved in the experiment to evaluate the immunogenicity of URRA individually and in composition with spherical particles (SPs). The control group (group 1) was not immunised. Other mice groups were immunised intramuscularly. Ten mice in each group were immunised once, and 15 mice were immunised twice, with a 21-day interval between immunisations. Mice in group 2 were immunised with 250 μg of SPs, in group 3 with 15 μg of URRA, and in group 4 with 250 μg of SPs and 15 μg of URRA. All samples were administered with PBS in a total volume of 0.26 mL. SPs, spherical particles obtained by the thermal remodelling of TMV; n, number of mice participating in the corresponding stage of the experiment; *, in group 4, only 14 mice were involved in the second immunisation, while nine mice were involved in blood sampling after the second immunisation.

**Figure 7 viruses-16-00438-f007:**
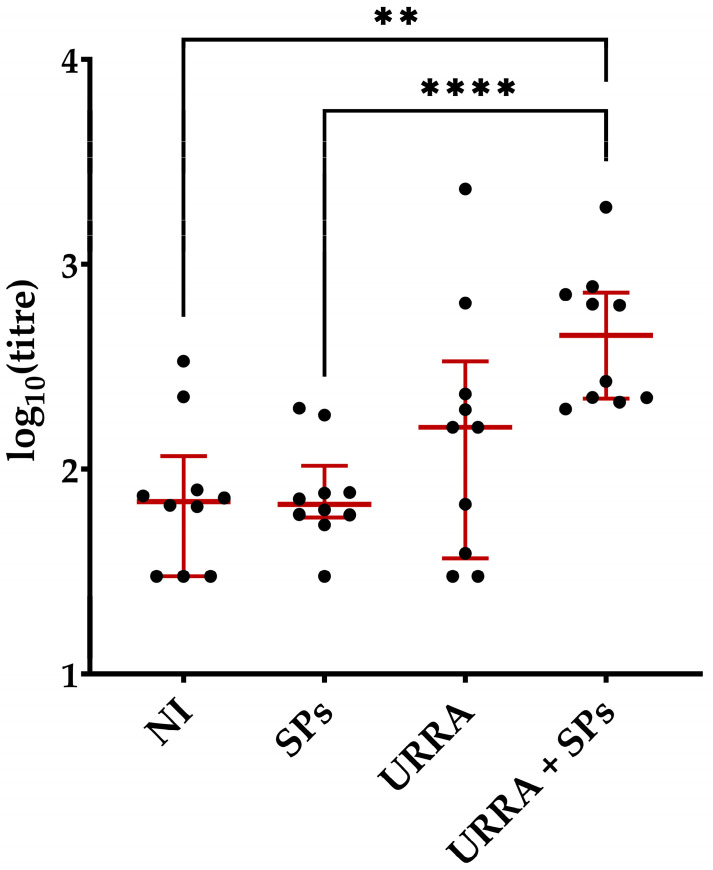
Immunogenicity of URRA individually and in composition with spherical particles (SPs) after the first immunisation. Anti-URRA IgG titres in four groups of mice are presented. The scheme of the study is presented in Figure 6. Sera titres were evaluated using indirect ELISA. The concentration of antigen used for adsorption on a microplate was 10 μg/mL. *p*-values were calculated using the Wilcoxon–Mann–Whitney Test with the Holm–Bonferroni correction. ●, IgG titres of individual mice; NI, non-immunised group; **, *p* < 0.01; ****, *p* < 0.0001; **—**, median. Error bars represent the interquartile range. Formulations used for the immunisation of corresponding groups of mice are marked under the horizontal axis. The complete data on anti-URRA sera titres for corresponding mice are presented in Appendix A.

**Figure 8 viruses-16-00438-f008:**
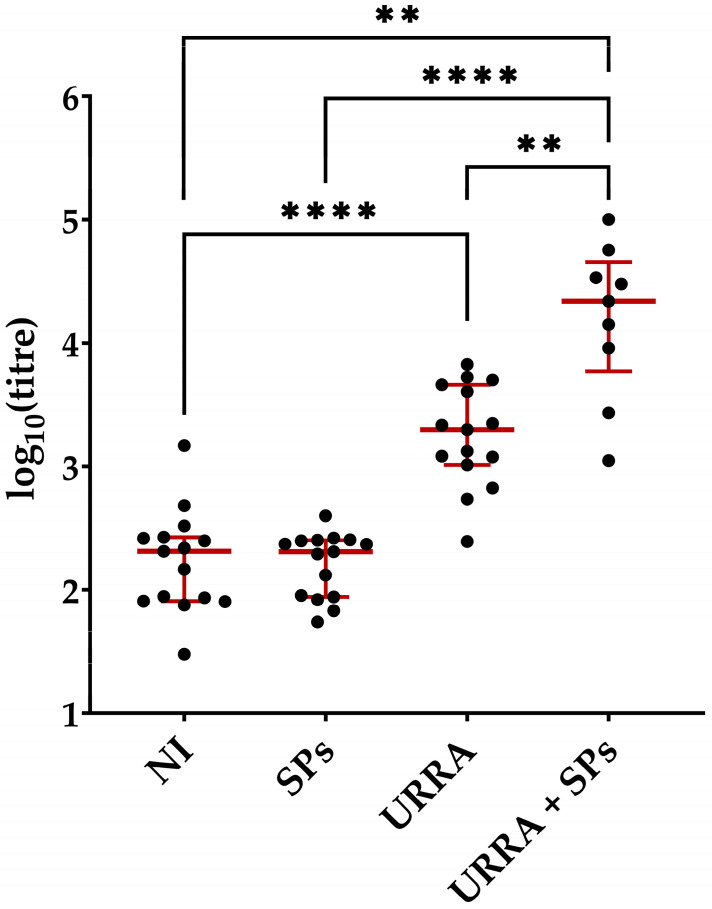
Immunogenicity of URRA individually and in composition with spherical particles (SPs) after the second immunisation. Anti-URRA IgG titres in four groups of mice are presented. The scheme of the study is presented in Figure 6. Sera titres were evaluated using indirect ELISA. The concentration of antigen used for adsorption on a microplate was 10 μg/mL. *p*-values were calculated using the Wilcoxon–Mann–Whitney Test with the Holm–Bonferroni correction. ●, IgG titres of individual mice; NI, non-immunised group; **, *p* < 0.01; ****, *p* < 0.0001; **—**, median. Error bars represent the interquartile range. Formulations used for the immunisation of corresponding groups of mice are marked under the horizontal axis. The complete data on anti-URRA sera titres for corresponding mice are presented in Appendix A.

**Figure 9 viruses-16-00438-f009:**
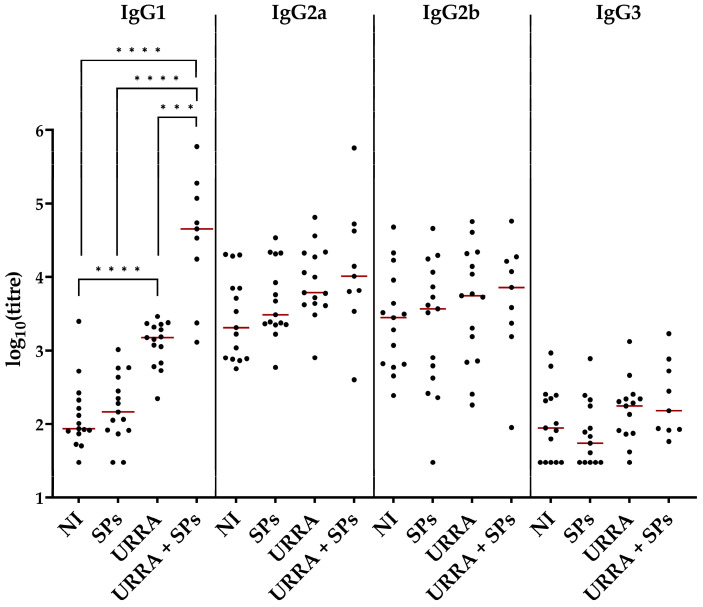
Comparison of anti-URRA IgG isotypes’ (IgG1, IgG2a, IgG2b, IgG3) titres in groups after the second immunisation. The scheme of the experiment is presented in Figure 6. Sera titres were evaluated using indirect ELISA. The concentration of antigen used for adsorption on a microplate was 10 μg/mL. *p*-values were calculated using the Wilcoxon–Mann–Whitney Test with the Holm–Bonferroni correction. ●, IgG isotype titres of individual mice; NI, non-immunised group; ***, *p* < 0.001; ****, *p* < 0.0001; **—**, median. Formulations used for the immunisation of corresponding groups of mice are marked under the horizontal axis. The IgG isotypes are marked above the corresponding graphs. The complete data on anti-URRA IgG1, IgG2a, IgG2b and IgG3 sera titres are presented in Appendix A, respectively.

**Figure 10 viruses-16-00438-f010:**
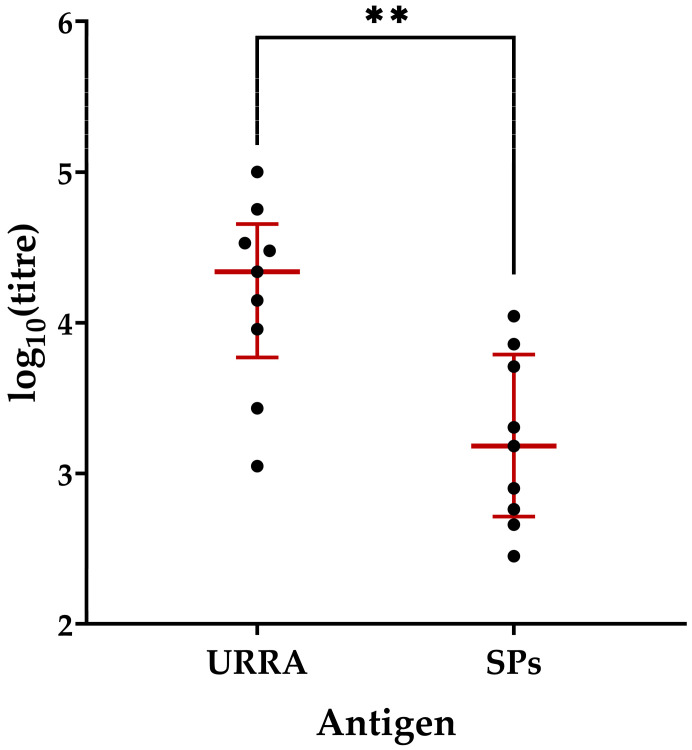
Comparison of anti-URRA and anti-SPs immune responses induced by the URRA + SPs composition after the second immunisation. The scheme of the study is presented in Figure 6. Sera titres were evaluated using indirect ELISA. The concentration of antigen used for adsorption on a microplate was 10 μg/mL. *p*-values were calculated using the Wilcoxon–Mann–Whitney Test. ●, IgG titres of individual mice; NI, non-immunised group; **, *p* < 0.01; **—**, median. Error bars represent the interquartile range. The complete data on anti-SPs sera titres for corresponding mice are presented in Appendix A.

## Data Availability

All the relevant data are provided in this paper and in Appendix A.

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
