# Peer review of "Novel Universal Recombinant Rotavirus A Vaccine Candidate: Evaluation of Immunological Properties"

_viruses, 2024, doi:10.3390/v16030438_

Round 1
Reviewer 1 Report
Comments and Suggestions for Authors
In this study, Ganovskiy and co-workers proposes a novel universal rotavirus vaccine candidate based on epitopes present in VP8* and immunological evaluation of the peptide in mice. The authors also evaluated immunogenicity of the universal recombinant rotavirus antigen (URRA) when associated with tobacco mosaic virus-based spherical particles (SPs). In the presence of the SPs, the immunogenicity of URRA was significantly increased, suggesting the potential of the SPs as an adjuvant. Overall, this is an interesting study providing evidence as a potential vaccine candidate. Although the immunogenicity of URRA was evaluated as well as the ability of the peptide to be recognised serologically by patient sera was established, the ability of the peptide to induce neutralising responses was not evaluated and would have further strengthened the study. The following revisions are proposed:
1. The introduction section of the manuscript is long and could be shortened.
2. “live-attenuated” vaccine – change throughout the manuscript
3. Lane 44: “the incidence of rotavirus-associated diarrhea is already high.”
4. The authors should clarify (expand) the statement made in lanes 47-48: “The extent to which existing live rotavirus vaccines provide cross-serotype protection is currently a controversial issue.
5. Also, lane 52 – the authors should provide references for the statement that recombinants /reassortants between rotavirus vaccine strains and field strains could be more pathogenic.
6. Lane 67: provide reference for “the main targets for neutralising antibodies”
7. In the introduction (lane 102) and discussion (lane 489) the authors speculate about the emergence of uncommon RVA strains, possibly as a result of vaccine pressure. The authors refer, amongst others, to a study by João et al 2020 who reported on the emergence of strains that were not detected prior to vaccine introduction. Mozambique uses Rotarix (G1P[8]) which one would expect a reduction in P[8]-containing strains. This was however not the case, with an increased detection of, amongst other, also G1P[8] strains. One should therefore be cautious not to oversimplify vaccine effectiveness as various, complex factors, such as malnutrition, contribute to the ability of vaccines to induce a protective immune response. Natural fluctuation in strain diversity should also not be excluded.
8. Lane 122: Was the entire overnight culture (3 mL) used to inoculate the 200 mL culture used for expression?
9. Lanes 145-148: Please rephrase – Both strains have been assigned as RVA by using PCR and ELISA (or similar).
10. Section 2.4: It is mentioned that plates were coated with URRA and SP for the detection of immune responses, the same antigens to raise immune responses. Ideally a different antigen than what was used to induce immune responses should be used for detection. In this instance, a G1P[8] rotavirus strain such as Wa could be used.
11. Lane 183: change “Than” to “Then”
12. Throughout the results section, please be careful not to repeat methodology unnecessary – examples, but not limited to: lanes 271-276; 354-360.
13. Figure 6: change description in the figure to: 1st/2nd/3rd immunisation day 0/21/42 (not 0/21/42 day)
14. Lane 431: The “level” (ratio?) of immune response to antigen……
Comments on the Quality of English LanguageSee above.
Author Response
We are grateful for close attention to our article and for the comments of the reviewer. The ability of URRA to induce neutralising responses is indeed important to be studied and we consider such experiments in our further research.
Here we provide point by point responses to the reviewer’s comments:
- The text of the “Introduction” section of the MS was edited.
- The text of the MS was edited.
- The text of the MS was edited according to the comment.
- The text of the MS was edited. The statement was extended, the references were added.
- The relevant references were added to the MS.
- The text of the MS was edited, the statement was removed.
- The text of the MS was edited, the potential impact of the natural evolutionary processes on the diversity of the RV strains after the introduction of vaccines in the region was discussed.
- Yes, 3 ml of the overnight culture (3 mL) were added to 200 ml of 2YT.
- The text of the MS was edited, the sentence was rephrased.
- The aim of the experiment was to evaluate the ability of SPs to increase the immunogenicity of URRA in the vaccine candidate. The protocol of the experiment was designed accordingly. Thus, the plates were coated with URRA in order to examine the titres of anti-URRA antibodies elicited. However, anti-G1P[8] titres elicited by URRA are interesting as well and might be evaluated in the further research.
- The text of the MS was edited.
- In these lanes we refer to the corresponding subsections of “Materials and Methods” section, since we used two different ELISA protocols and sera from two different sources in the research, and we found it necessary to clarify which protocol and which sera we used in each case. The lack of this information might lead to confusion in the reader. The rest of text of the MS was checked, several excessive repeats of methodology were removed.
- Figure 6 was edited.
- The text of the MS was edited.
Reviewer 2 Report
Comments and Suggestions for Authors
Using plant viruses in biotechnology and particularly in vaccine design is a rapidly growing area of research. This article is devoted to the development of the rotavirus vaccine based on the universal rotavirus antigen URRA and plant virus-derived spherical particles. URRA was shown to interact with the sera to two relevant rotavirus strains and to be immunogenic when co-administered with spherical particles obtained from TMV. The combination of using the conservative epitope and extended consensus of VP4 protein combined with a proposed adjuvant is a promising approach to the rotavirus vaccine design. The results obtained in the present work are interesting and convincing. The use of the sera to currently circulating rotavirus strains is the main strength of the study. As a logical continuation of the in vitro research I recommend to evaluate the interaction of anti-URRA sera with the rotavirus strains described in the paper. My point by point comments are listed below. These remarks do not diminish the overall worth of the current research.
1. Line 555: “titers”; Line 556: “immunization” x2. Must be edited according to the style of English language used in the rest of the paper.
2. Recombinant proteins produced in E. coli expression system are commonly contaminated with LPS. In this regard, did you perform any endotoxin assays with samples used for immunogenicity studies?
Comments on the Quality of English LanguageLine 555: “titers”; Line 556: “immunization” x2. Must be edited according to the style of English language used in the rest of the paper.
Author Response
We appreciate the positive feedback of the reviewer. Evaluation of the ability of anti-URRA sera to interact with the rotavirus strains described in the paper is in our plans for further research, thanks for the suggestion.
Here we provide point by point responses to the reviewer’s comments:
- The text of the MS was edited. Typos were corrected, the rest of the text was checked.
- The LPS contents in the samples used for immunization of the groups 2 (SPs), group 3 (URRA) and group 4 (URRA + SPs) were measured using EC Endotoxin Test Kit (End-point Chromogenic Assay, EC64405S) (Bioendo™, China). Endotoxin contents in each of three formulations were less than 0,5 EU/dose (the dose is 260 μl per animal).
Comments on the Quality of English Language
- The text of the MS was edited. Typos were corrected, the rest of the text was checked.